# The potential of chemical bonding to design crystallization and vitrification kinetics

Christoph Persch [1], Maximilian J. Müller [1], Aakash Yadav [1], Julian Pries [1], Natalie Honné[1], Peter Kerres[1], Shuai Wei [1,2], Hajime Tanaka [3,4], Paolo Fantini[5], Enrico Varesi[5], Fabio Pellizzer[6] & Matthias Wuttig [1,7,8✉]

Controlling a state of material between its crystalline and glassy phase has fostered many real-world applications. Nevertheless, design rules for crystallization and vitrification kinetics still lack predictive power. Here, we identify stoichiometry trends for these processes in phase change materials, i.e. along the GeTe-GeSe, GeTe-SnTe, and GeTe-Sb$_2$Te$_3$ pseudo-binary lines employing a pump-probe laser setup and calorimetry. We discover a clear stoichiometry dependence of crystallization speed along a line connecting regions characterized by two fundamental bonding types, metallic and covalent bonding. Increasing covalency slows down crystallization by six orders of magnitude and promotes vitrification. The stoichiometry dependence is correlated with material properties, such as the optical properties of the crystalline phase and a bond indicator, the number of electrons shared between adjacent atoms. A quantum-chemical map explains these trends and provides a blueprint to design crystallization kinetics.

---

[1] I. Institute of Physics, Physics of Novel Materials, RWTH Aachen University, Aachen, Germany. [2] Department of Chemistry, Aarhus University, Aarhus C, Denmark. [3] Institute of Industrial Science, University of Tokyo, Meguro-ku, Tokyo, Japan. [4] Research Center for Advanced Science and Technology, University of Tokyo, Meguro-ku, Tokyo, Japan. [5] Micron Technology Inc., Vimercate, Italy. [6] Micron Technology Inc., Boise, ID, USA. [7] Jülich-Aachen Research Alliance (JARA FIT and JARA HPC), RWTH Aachen University, Aachen, Germany. [8] PGI 10 (Green IT), Forschungszentrum Jülich GmbH, Jülich, Germany. ✉email: wuttig@physik.rwth-aachen.de

Upon cooling down, a supercooled liquid either crystallizes or forms a glass. Both crystallization and vitrification are topics of profound technological significance and provide exciting scientific challenges[1]. These processes are of economic importance as exemplified by the relevance of producing polymer plastics, growing single crystals for (opto-)electronics, the hardening of steel[1] as well as making chocolate with the right consistency[2]. Hence, it would be highly desirable to design crystallization and vitrification kinetics[3]. The glass-forming ability has received much attention in the last two decades with the advent of bulk metallic glasses[4,5] and phase change materials employing the glass-crystal transition[6]. Theoretically, vitrification and crystallization were mostly discussed independently from each other, but recent studies have revealed the fundamental link between the two phenomena[3,7–9]. These studies have lately even been extended to electronic system[10,11] ('charge liquids').

One possible way to achieve the goal of tailored crystallization and vitrification kinetics would be a fundamental understanding of how the kinetics of crystallization and vitrification depend on the chemical bonding between the constituent atoms. Interestingly, there are remarkable differences in the vitrification of iono-covalent systems like $SiO_2$ and metallic systems such as simple elemental metals and metallic alloys[9,12] brass and bronze. While $SiO_2$ is a good glass former, crystallization is so fast in metals that it is very challenging to produce metallic glasses. Only very recently has it become possible to reach the huge cooling rates needed to create a metallic glass even from an elemental metal[13].

These differences in crystallization and vitrification between metallic and covalently bonded systems must be closely related to differences in the interaction between the atoms involved. Materials can be classified into two groups in terms of their main driving force to order: directional-bonding-dominated and entropy-dominated systems[14]. Metals may be categorized into the latter since atomic packing entropy plays a significant role in structural ordering. On the contrary, covalently bonded materials are classified into the former. It is interesting to note that in metallic bonding, electron delocalization is the mechanism responsible for the energy minimization, whereas, in covalent bonding, electron localization between adjacent atoms enables energy minimization. Recently, phase change materials have been identified as a class of materials whose crystalline states have a bonding mechanism different from metallic and covalent bonding types[15–17]. This immediately raises an intriguing question of whether crystallization of phase change materials shows differences from the crystallization of solids that utilize covalent or metallic bonding. This question is not only of academic interest but also relevant for applications. Phase change materials like GeTe or $Ge_2Sb_2Te_5$ are characterized by pronounced differences of optical and electrical properties between their amorphous and crystalline states[18]. Their ability to be rapidly switched between the two states is employed in several applications ranging from optical to electronic data storage, neuromorphic computing and active photonic devices[18–26]. The design of their crystallization and vitrification kinetics therefore holds important opportunities to improve their application potential. This goal has motivated numerous research activities to unravel the origin of fast crystallization in PCMs. Previous studies have emphasized the significance of fourfold rings[27–31] in the amorphous (glassy) phase and have elucidated the role of reduced stochasticity on crystal nucleation[32].

Here, a different approach is employed. We explore the impact of systematic changes in chemical bonding on crystallization kinetics. To do so, we investigate fast crystallization in GeTe, a prototypical phase change material, and alloy it with GeSe, SnTe, and $Sb_2Te_3$, respectively. GeSe is significantly more covalent than GeTe, while SnTe is characterized by a larger charge transfer. All three material systems possess well miscible liquid phases[33]. They are characterized by an octahedral-like atomic arrangement in their crystalline phases, albeit with different levels of distortions[34–36], characteristic for p-bonded compounds. We thus avoid mixing GeTe with tetrahedral, $sp^3$-bonded semiconductors. Hence, potential differences in crystallization and vitrification kinetics should be governed by differences in chemical bonding, as will be proven below.

## Results

**Minimum time for crystallization.** Utilizing an optical tester, we can measure the minimum time for crystallization $\tau$ as follows. The as-deposited amorphous sample is exposed to laser pulses of varying power and length to determine the onset of crystallization. These laser pulses drastically change the sample reflectance upon crystallization for typical phase change materials, since the dielectric function $\varepsilon(\omega)$ differs significantly in the amorphous and crystalline states[37,38] (see Supplementary Table 1). This enables the determination of $\tau$, which changes systematically with stoichiometry (as shown in Fig. 1, Supplementary Table 1, and Supplementary Figs. 8 and 9). Note, that the minimum crystallization time $\tau$ measured here includes the incubation time of crystal formation. This causes the measured values of $\tau$ to be longer than what is expected for actual PCM-devices[39], e.g., when sub-critical nuclei are frozen in during melt-quenching, accelerating the nucleation stage[40,41]. The pros and cons of studying crystallization of the melt-quenched state vs the as-deposited amorphous state are discussed in the supplement.

Going from as-deposited GeTe (620 ns) to $GeTe_{0.8}Se_{0.2}$ ($9.6 \times 10^3$ ns) and $GeTe_{0.6}Se_{0.4}$ ($4.1 \times 10^4$ ns) leads to a significant increase in minimum crystallization time. For $GeTe_{0.4}Se_{0.6}$ an increase to $1.3 \times 10^6$ ns is found, while crystallization even takes $8.7 \times 10^6$ ns for $GeTe_{0.3}Se_{0.7}$, see Fig. 1. It is striking that isoelectronic replacement of Te by Se, i.e. a replacement by a chemically similar element, leads to such a pronounced increase by a factor of $10^4$. On the other hand, alloying GeTe with SnTe leads to a significant reduction in crystallization time. $Ge_{0.8}Sn_{0.2}Te$ already crystallizes in 250 ns, $Ge_{0.6}Sn_{0.4}Te$ even switches in 80 ns and $Ge_{0.5}Sn_{0.5}Te$ crystallizes in only 25 ns. How can these dramatic changes in crystallization speed be explained?

Interestingly, the increasing minimum time for crystallization $\tau$ is accompanied by a simultaneous decrease in the reflectance of the crystalline film, as shown in Fig. 2b. In contrast, the correlation between the reflectance of the amorphous film before crystallization and the minimum time for crystallization $\tau$ is less evident, as can be seen in Supplementary Fig. 3. This is surprising since crystallization should depend on the properties of both the amorphous and crystalline states. Nevertheless, Fig. 2 and Supplementary Fig. 3 imply that the crystalline state and its electronic structure, which govern optical properties seem to have a dominant impact on crystallization for phase change materials. How can this finding be rationalized?

The optical properties of solids, i.e. their dielectric function $\varepsilon(\omega)$ are closely related to the nature of the electronic states in the vicinity of the Fermi level $E_F$, i.e. valence (VB) and conduction band (CB) states. The states below $E_F$ also constitute the chemical bonds. Crystalline phase change materials possess unconventional VB and CB states, as discussed next[38,42,43]. Both VB and CB states in crystalline PCMs are dominated by p-electrons, which form a σ–bond. Phase change materials like GeTe only possess on average three p-electrons per atom forming their valence bands, but have an octahedral-like atomic arrangement[44] (see Supplementary Fig. 2). Hence, for each of the six neighboring atoms, only a single electron is available to create a bond. This leads to a unique situation, where adjacent atoms are held together by a

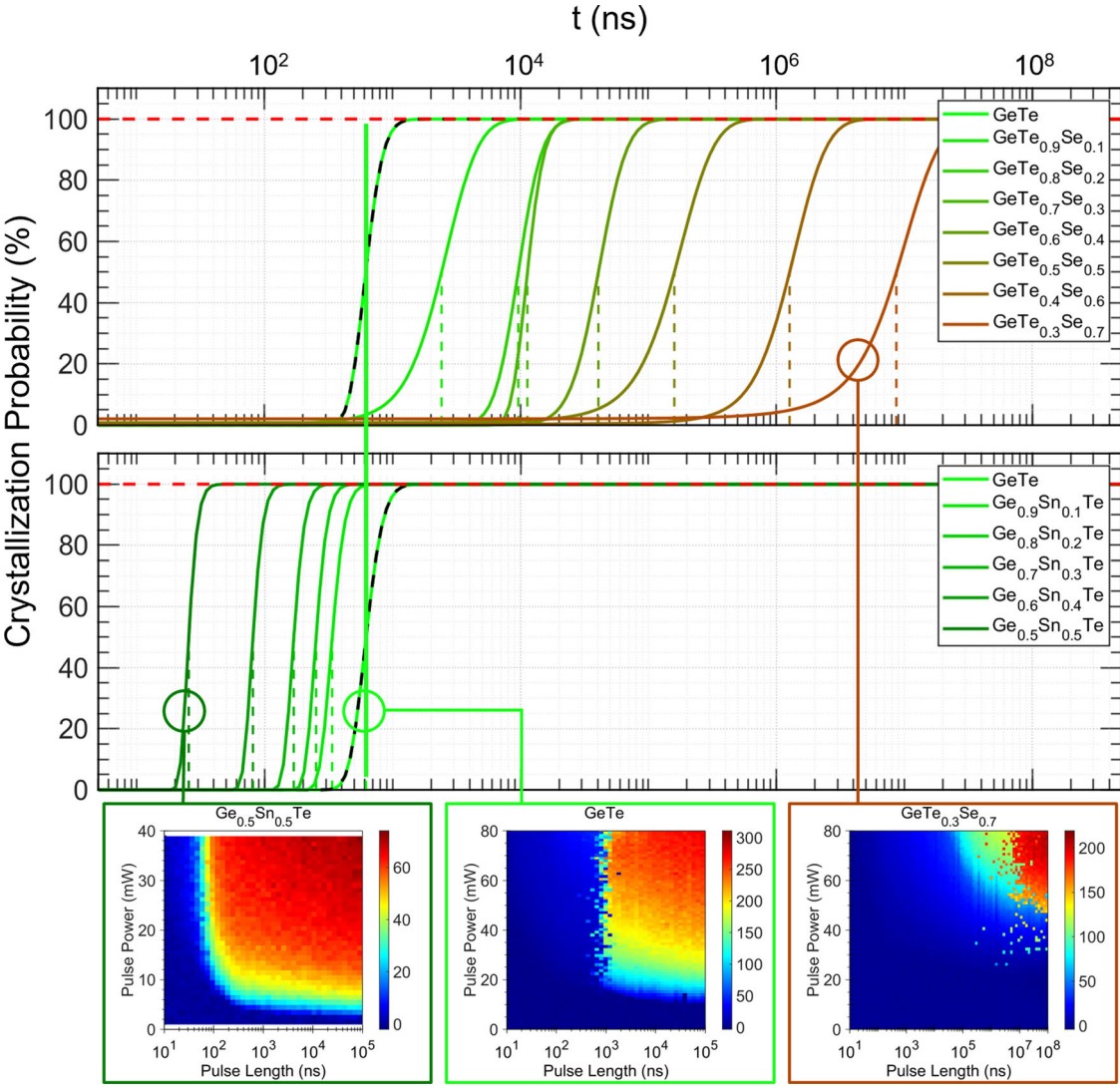

**Fig. 1 Crystallization speed of different chalcogenides.** Minimum crystallization time as a function of laser pulse length for the GeTe-GeSe and GeTe-SnTe pseudo-binary lines. While alloying GeTe with GeSe increases the crystallization time by several orders of magnitude, mixing GeTe with SnTe reduces the crystallization time significantly. The solid lines are fits to the data (see SI for additional information), while the vertical dashed lines represent the minimum crystallization time deduced. Experimental data revealing the effect of laser pulses of different length and power for three different compounds are shown beneath (see Methods section). The color bars denote the change of sample reflectance upon crystallization (in percentage).

single electron, distinctively different from the two electrons which form an electron pair in covalent bonds[45]. This conclusion is supported by quantum chemical calculations which yield the number of electrons shared (ES) between neighboring atoms[17]. Indeed, compounds with an octahedral atomic arrangement such as SnTe (at room temperature) are characterized by about one electron shared between adjacent atoms. This has pronounced consequences for the optical properties since the states responsible for the optical transitions between the valence and conduction band are governed by p-orbitals. The similarity of the wave functions for the valence and conduction band states leads to a large matrix element for interband transitions responsible for the high reflectance depicted in Fig. 2c. Yet, GeTe is characterized by a small distortion away from the perfect octahedral arrangement (Peierls distortion). This increases the bandgap and the number of electrons shared between adjacent atoms to ~1.1[34], reducing the optical reflectance as sketched in Supplementary Fig. 2. Thus, the large sample reflectance is due to the nature of the electronic states in the vicinity of the Fermi level. Therefore, the chemical bond in the crystalline state (as

characterized by ES) and the film's reflectance are closely interwoven. Reducing the octahedral distortion reduces the number of electrons shared, yet increases the matrix element for the optical transitions and hence the sample reflectance. This apparently causes a pronounced decrease in the minimum time for crystallization $\tau$.

These findings provide two guidelines on how to identify materials with ultra-fast crystallization. We can either experimentally search for compounds with octahedral-like atomic arrangement, yet small distortions and an average of 3 p-electrons per atom, or perform quantum chemical calculations and search for compounds with near-perfect octahedral arrangement, which share about 1 electron between adjacent atoms. Before discussing the correlation between the minimum crystallization time and chemical bonding further, we explore the process competing with crystallization, i.e. vitrification.

**Vitrification.** Upon cooling down a liquid, the material will either crystallize or form a glassy state. For applications of phase change materials, it is desirable to realize rapid crystallization. However,

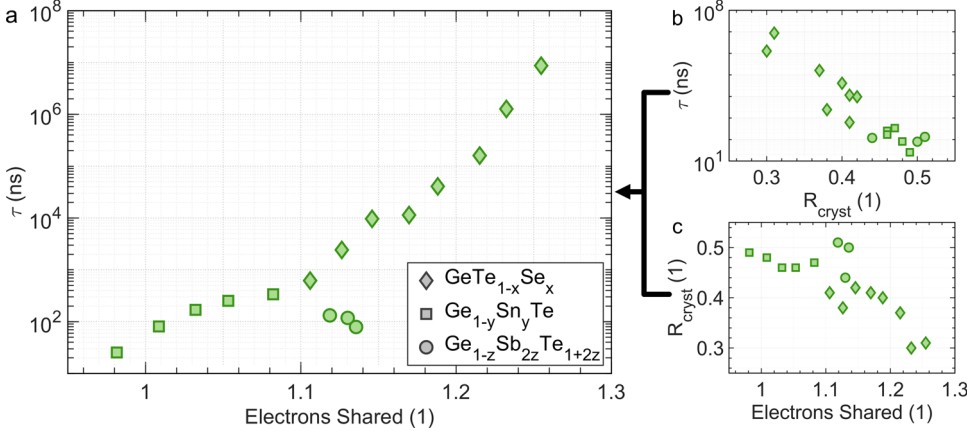

**Fig. 2 Minimum crystallization time $\tau$. a** Dependence of the crystallization time $\tau$ on the number of electrons shared (ES) between adjacent atoms in the crystalline phase. A pronounced increase of $\tau$ with the number of electrons shared between adjacent atoms is found. This clear trend is related to a systematic change of the reflectance of the crystalline samples. The crystallization time increases upon decreasing reflectance of the crystalline sample (**b**), which is due to the close link between the number of electrons shared between adjacent atoms and the optical properties of the crystalline sample (as discussed in Supplementary section IV in more detail).

it is also mandatory to have a reasonably stable glassy (amorphous) state. If the undercooled liquid cannot be vitrified upon quenching from the melt because of crystallization, then re-amorphization of the material is impossible precluding applications as a phase change material. A key quantity to characterize the ease of glass formation is the reduced glass transition temperature $T_{rg} = T_g/T_m$, which characterizes the transition from the glassy to the undercooled liquid state and hence does not involve crystallization. As shown by Turnbull[9], the higher the value of $T_{rg}$, the easier the material forms a glass. Good glass formers like $GeO_2$, $SiO_2$, and $B_2O_3$ are characterized by $T_{rg}$ values of 0.59, 0.75, and 0.74, respectively[46], while many poor glass formers, e.g. the elemental metals Ag, Pt, Ta, and Fe, possess $T_{rg}$ values between 0.3 and 0.4[47]. This raises the question of how the reduced glass transition temperature $T_{rg}$ changes with stoichiometry. Good phase change materials, such as compounds on the pseudo-binary line between GeTe and $Sb_2Te_3$, are rather poor glass formers, and thus, it is challenging to measure their glass transition temperature[48]. This is confirmed by calorimetric measurements, summarized in Supplementary Fig. 4. Only the five most GeSe-rich compounds on the GeSe-GeTe pseudo-binary line show a clear glass transition at heating rates close to the standard heating rate $\vartheta_s$ of 20 K/min, enabling the determination of $T_g$ (Supplementary Fig. 5, triangles pointing up). For all other compounds of this system, as well as samples from the GeTe-SnTe and GeTe-$Sb_2Te_3$ pseudo-binary lines, the glass transition is obscured by fast crystallization.

However, a good estimate of the glass transition temperature is necessary to investigate stoichiometry trends. Upon increasing the heating rate, the crystallization temperature increases more rapidly than the glass transition temperature[49], revealing more of the glass transition. Still, raising the heating rate up to 60,000 K/min does not expose the glass transition for all compounds investigated. Instead, the onset temperature of the glass transition $T_o$ can be determined as explained in Supplementary Fig. 4. As shown in Supplementary Fig. 5 the reduced endothermic onset temperature $T_{ro}$ displays the same dependence on the number of electrons shared (ES) as $T_{rg}$ (Supplementary Fig. 5, triangles pointing up and down, present data and literature values[49,50], respectively) but is shifted to higher values. Hence, $T_{ro}$ has a similar stoichiometry dependence as $T_{rg}$ but provides a larger data density. As depicted in Fig. 3a, the value for $T_{ro}$ is much higher for GeSe (0.605) than the corresponding value for GeTe

(0.497). Furthermore, the Se-rich compounds are characterized by relatively constant $T_{ro}$ values. For these compounds even laser pulses of 0.1 s did not produce a clear change in optical reflectance, showing that these compounds are not suitable as phase change materials. Furthermore, these GeSe-rich compositions showed a much smaller optical reflectance in the crystalline state, indicative for covalent bonding[51]. Reducing the Se-content leads to a drastic decrease in $T_{ro}$ and much faster crystallization. In this stoichiometry range, we observe a linear decrease in $T_{ro}$ and an exponential decrease in $\tau$, see Fig. 3a, b. Hence, replacing Se by Te leads to much faster crystallization, but also destabilizes the amorphous phase and thus hampers glass formation.

It is surprising that Fig. 3 seems to relate the bonding descriptors of the crystalline phase, i.e. the ES value for this phase, with the stability of the glassy phase, i.e. $T_{ro}$ and $T_{rg}$, even though bonding and properties of the amorphous phases are significantly different from their crystalline counterpart[37,52–54]. For example, in GeTe, the Peierls distortion is much larger in the amorphous than the crystalline phase[38]. Such an increase in the Peierls distortion leads to an increase in the number of electrons shared between adjacent atoms, which increases the bond strength[53]. Hence, it is highly desirable to determine the number of electrons shared between adjacent atoms (ES) for amorphous (glassy) phase change materials, too. While the large system sizes required for these quantum chemical calculations of the amorphous phases are demanding, these computations could unravel the correlation between the glass-forming ability and chemical-bonding character, as well as their link to the atomic arrangement in the glassy state, a fascinating perspective. In the last decade, significant work has been performed to study the atomic arrangement of amorphous PCMs. Besides emphasizing considerable differences in the atomic arrangement between amorphous and crystalline phases, compelling evidence for fourfold rings as a prerequisite to nucleation has been found[27–31]. Such fourfold rings describe symmetric configurations where all nearest neighbor distances within the ring are similar, and hence locally the ES value could be close to 1. Such a situation appears to be very favorable for rapid crystallization, in line with the trend shown in Fig. 3a.

The close link between crystallization (i.e. $\tau$) and vitrification (i.e. $T_{rg}$ and $T_{ro}$) for the chalcogenides studied here is depicted in Fig. 3b. For the three pseudo-binary lines investigated a clear correlation is found, where an increase of glass-forming ability ($T_{ro}$) is accompanied by a concomitant increase of the minimum

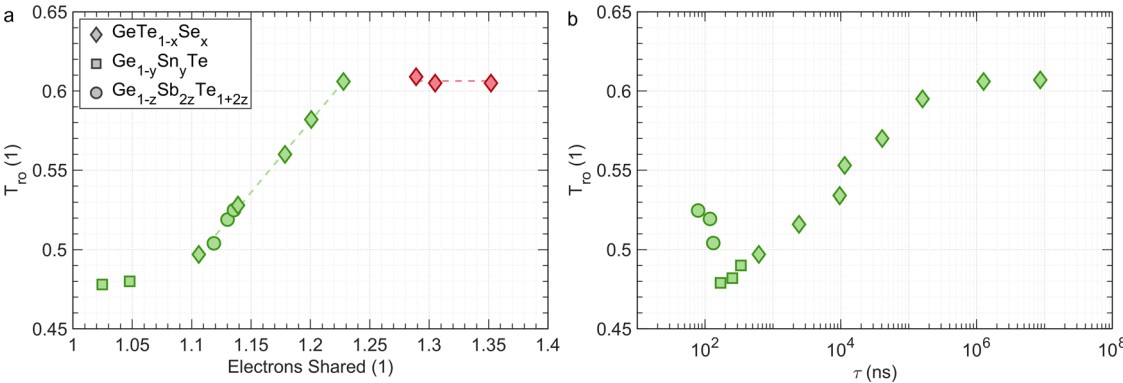

**Fig. 3 Glass transition. a** Dependence of the reduced onset temperature $T_{ro}$ for glass formation upon ES (for the crystalline phase). $T_{ro}$ was measured at a heating rate of 60,000 K/min. Replacing Te in GeTe by Se leads to a significant increase in glass-forming ability as characterized by $T_{ro}$. **b** Correlation between the minimum time of crystallization $\tau$ and $T_{ro}$. Ultrafast crystallization (low values of $\tau$) is accompanied by low glass stability (small values of $T_{ro}$). Interestingly, compounds on the pseudo-binary line between GeTe and $Sb_2Te_3$, such as $Ge_2Sb_2Te_5$ are characterized by ultrafast crystallization, but slightly improved stability of the glassy state. $T_{ro}$ data are extrapolated from powder samples (see Methods section).

crystallization time $\tau$. The relationship between vitrification and crystallization has already been considered more than 50 years ago[9], but experimental data discussing the stoichiometry dependence and the impact of chemical bonding are sparse. The data displayed in Fig. 3b emphasize the close relationship between crystallization and vitrification for several chalcogenides. For the application of PCMs, this is a challenge since it reveals that increased switching speeds are accompanied by a reduced stability of the amorphous phase. Fortunately, Fig. 3b also shows some chalcogenides that crystallize very rapidly, but have a more stable glassy phase, i.e. higher $T_{ro}$ and $T_{rg}$, respectively. This increased stability of the amorphous phase is observed for compounds on the pseudo-binary line between GeTe and $Sb_2Te_3$, such as $Ge_2Sb_2Te_5$. These compounds are prototype phase change materials employed in optical and electronic data storage[18,24–26,55]. Figure 3b helps to understand their industrial relevance. It would be rewarding to explain the increased glass-forming ability of $Ge_2Sb_2Te_5$ compared to GeTe, which might be related to a more pronounced Peierls distortion of amorphous $Ge_2Sb_2Te_5$ as discussed in more detail in the supplement.

## Discussion

Crystallization kinetics have been long known to differ for metals and covalently bonded solids. Recently, a map has been devised, which separates ionic, covalent, and metallic bonding based on electrons transferred and shared between adjacent atoms[17]. These two quantities can be determined in quantum chemical calculations[56]. The chalcogenides studied here are located in a well-defined region of this map between metallic, covalent, and ionic types of bonding. This region characterizes a distinct bond, different from metallic, ionic, and covalent bonding as can be seen from characteristic differences in material properties[16] and bond rupture[15]. This type of bond has been coined metavalent bonding[16] (MVB). The 2D map to separate bonding mechanisms can be extended to a 3D property map, if a property such as the minimum crystallization time $\tau$ or the glass-forming ability, described by $T_{ro}$ or $T_{rg}$ is chosen as the z-axis. Such maps are shown in Fig. 4a, b.

Interestingly, systematic trends for crystallization and vitrification are shown in Fig. 4. While the crystallization speed increases tremendously in the region between covalent and metallic bonding, i.e. for metavalently bonded solids, simultaneously the glass-forming ability decreases drastically. Apparently, these trends are governed by the decreasing number of electrons shared between adjacent atoms (ES). It has been demonstrated in Fig. 2 that the increasing crystallization speed is accompanied by an increasing optical reflectance, indicative of a decreasing size of the Peierls distortion. Presumably, this leads to weaker bonds, in line with lower ES values, as well as the softening of transverse optical phonons and an increased anharmonicity, evidenced by high values of the Grüneisen parameter for transverse optical modes[16]. These weaker bonds facilitate atomic rearrangements.

The findings presented here provide a blueprint to tailor crystallization and vitrification for chalcogenides and enable surprising predictions. However, there are still exciting open questions, which address very different aspects of crystallization and vitrification kinetics. In theories discussing crystallization kinetics[9], the three fundamental quantities are the heat of fusion (crystallization), i.e. the thermodynamic driving force for crystallization, the interfacial tension between crystal and glass, a measure of the similarity of both phases[7,8] as well as the atomic mobility in the undercooled liquid phase. We have revealed that the reduction of the octahedral distortion in a crystal is related to a decrease in the crystallization time $\tau$ and glass-forming ability. This suggests that properties of the crystalline phase are related with the glass-forming ability, i.e. the transition between the glassy and the undercooled liquid state. How can we understand this? It has recently been shown for simple model liquids that the similarity between preordering in melt and the crystal structure is a critical factor reducing the crystal-liquid interfacial tension and, accordingly, the glass-forming ability[7,8]. Yet, for the phase change materials discussed here, there is clear evidence that the atomic arrangement in the crystalline, amorphous, and liquid state differs[27–31,57]. Smaller differences in atomic arrangement and properties exist between these phases in covalently bonded chalcogenides such as GeSe[51]. Yet, GeTe crystallizes much more rapidly than GeSe. This implies that the interfacial energy for GeTe is rather small even though the atomic arrangement changes significantly at the interface between the amorphous and the crystalline state[58]. This unconventional behavior can be attributed to a peculiarity of the energy landscape of the crystalline phase of metavalently bonded solids. Since metavalent bonding is characterized by a competition between electron localization and delocalization, a change of atomic arrangement, such as the size of the Peierls distortion does not require a large energy[34], leading to low interfacial energies even for markedly different atomic arrangements in both phases. We hence speculate that octahedral-like local structures formed as locally

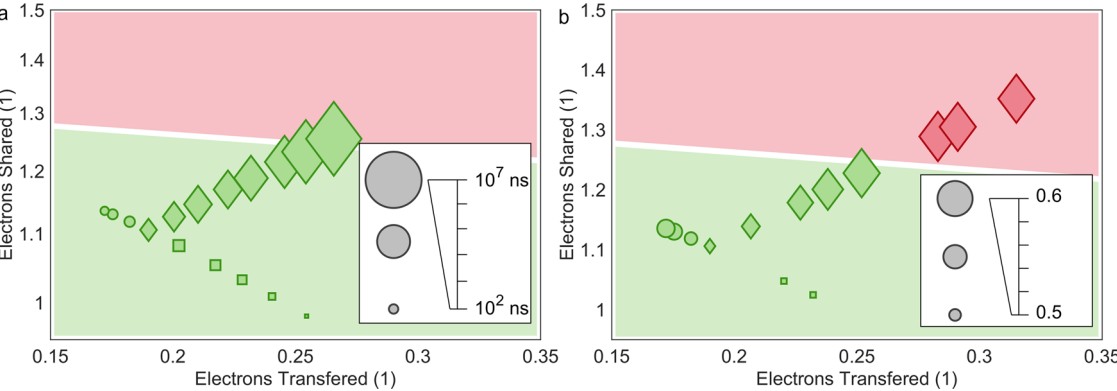

**Fig. 4 Change of kinetics with bond indicators.** Dependence of the minimum crystallization time $\tau$ (**a**) and the reduced onset temperature $T_{ro}$ for glass formation (**b**) upon two chemical bond quantifiers, the number of electrons transferred and shared between adjacent atoms. A pronounced decrease of the minimum time to crystallize and the glass-forming ability is observed in the metavalent bonding region (green background) between covalent (red) and metallic bonding. Figure 4a, in conjunction with Fig. 2a, shows that the crystallization time $\tau$ is clearly correlated with the number of electrons shared. For the Se-rich compounds, which are covalently bonded such as GeSe, crystallization is not even discernible in the optical tester.

favored structures in a supercooled liquid state act as crystal precursors, which compete with more distorted local structures that interfere crystallization. Although this argument should be confirmed, the crucial role of octahedral structures as precursors of crystalline ordering is consistent with previous findings[27–31] that a higher fraction of fourfold rings leads to faster crystallization. It also seems reasonable that the fraction of weakly distorted octahedral structures in a supercooled liquid state is higher in lower ES systems. Indeed, it has recently been shown that the undercooled liquid of phase change materials like $Ge_{15}Sb_{85}$ and $Ag_4In_3Sb_{67}Te_{26}$ are characterized by a liquid–liquid phase transition, which marks the onset of Peierls distortions at lower temperatures[59]. Presumably this phase transition provides the fast atomic rearrangement required for fast crystallization at elevated temperatures, yet stability against recrystallization at lower temperatures.

It would thus be highly desirable to produce maps, but now depicting systematic trends for the heat of fusion (crystallization) and the interfacial tension as a function of ES and ET. It would then be good to produce ES and ET values for the glassy phases of the compounds studied here and other chalcogenide glasses. Finally, it should be rewarding to extend the use of the chemical bond quantifiers, i.e. ES and ET, to describe crystallization and vitrification kinetics in other classes of materials. Are there similar trends also for the tetrahedrally coordinated, covalent systems and metals, i.e. can we predict their vitrification and crystallization behavior based on quantitative data for the number of electrons shared and transferred? If this goal could be reached, this would provide a new scientific perspective on the art of making glasses and producing crystals.

## Methods

**Sample preparation and layout.** $GeTe_{1-x}Se_{x}$- and $Ge_{1-y}Sn_{y}Te$-samples have been prepared via DC-magnetron sputter deposition from stoichiometric GeTe-, GeSe-, and SnTe-targets. To achieve chalcogenide compounds on the pseudo-binary lines between the end members, two magnetrons were operated simultaneously, while the substrates where rotated. To avoid formation of superlattice-like structures and to ensure good intermixing, the power of the magnetrons was set sufficiently low. For deposition, an Ar-flow of 60 sccm was used, resulting in a pressure of $1.3 \times 10^{-2}$ mbar. GST-samples were deposited from stoichiometric targets. EDX-measurements were carried out to verify the film stoichiometry.

To study the crystallization speed kinetics, a 30 nm thick layer of the chalcogenide is sandwiched between a 10 nm bottom and a 100 nm top layer of $(ZnS)_{80}(SiO_2)_{20}$ deposited by RF-magnetron sputtering on Si(100) substrates without breaking the vacuum. (F)DSC measurements were performed on powder samples obtained from depositing films of ~6 μm on thin steel sheets, the material was grounded into a fine powder after removal from the substrate.

Samples used for measuring the crystallization speed and samples used for measuring calorimetric quantities were prepared in separate preparation campaigns tailored to the specific requirements. Thus, stoichiometries of the two samples sets are not identical (compare Supplementary Table 1 for details).

**Minimum time for crystallization $\tau$.** The minimum time for crystallization $\tau$ was measured with a pump-probe setup using two lasers. The wavelength of the pump and probe are 658 nm and 639 nm, respectively. Both beams are combined via an optical fiber forming a spot of 2.3 μm ($1/e^2$) in diameter on the sample, measurements have been performed at 30 °C in a low-pressure Argon-atmosphere. Pump pulses were applied to locally heat the sample. The probe laser measured the change in reflectance after each single pump pulse compared to the reflectance before the pump pulse, providing a PTE-diagram, which depicts the effect ($E$) of different pulse power ($P$) between 0 and 90 mW and pulse lengths ($T$) between $10^1$ ns and $10^8$ ns (see Supplementary Fig. 6). The maximum laser power corresponds to a power density of $4.3 \times 10^6$ W/cm$^2$ in the center of the beam.

**Calorimetric measurements.** Calorimetric data was obtained using a PerkinElmer Diamond DSC and a Mettler Toledo Flash DSC 1 (FDSC). The excess heat capacity data was obtained by subtracting the rescan of the crystallized specimen. Data obtained by FDSC was additionally normalized using the enthalpy of crystallization determined at 20 K/min in DSC. Whenever possible, the glass transition temperature was obtained from conventional DSC by an onset construction.

In FDSC at 60,000 K/min, the exothermic enthalpy release due to structural relaxation ceases when the glass transition is about to occur, enabling the determination of the endothermic onset temperature $T_o$ as exemplified in Supplementary Fig. 4. The temperature scales of both DSC and FDSC was calibrated for each heating rate by the onset temperature of melting of pure Indium.

To evaluate calorimetric measurements together with crystallization measurements, $T_{ro}$-values fitting the composition of the crystallization samples were linear extrapolated from the two nearest powder samples. This procedure was conducted for all samples which were produced by simultaneous sputter deposition from two targets.

**Bond characteristics ET and ES.** Quantum chemical calculations were utilized to determine the two bond indicators. Additional information on the calculation can be found in[17]. The computations utilize a program developed by Golub and Baranov[56], which determines the formation of electron pairs between adjacent atoms and their electron transfer. The latter quantity is defined as the charge in the (Bader) basin surrounding an atom minus the number of electrons of the corresponding free atom. To facilitate the comparison of different compounds we divide the electron transfer by the formal oxidation state (ET).

The number of electron pairs formed between adjacent atoms characterizes the bond between a pair of atoms. Yet, typically an atom has several neighbors. One hence could focus on the first three nearest neighbors, determine the corresponding values for the number of electron pairs formed between these pairs of atoms and calculate the average value. Instead, we consider all neighbors and determine an average which weighs the distance of the atoms as explained in the supplement. The resulting number is multiplied by two, hence we are not determining the number of electron pairs formed between two atoms, but the number of electrons shared between them (ES). Comparing the results of quantum chemical calculations using two different software packages (dGrid and critic2) produces very similar results for ES and ET, showing that these two numbers are indeed

reliable chemical bond indicators. Hence, we obtain well-defined values for ES and ET for different end members (GeTe, GeSe, and SnTe). Since all these compounds are miscible and form solid solutions, which follow Vegard's law for their lattice constants for each of their solid phases, we also use this law to derive the composition averaged value for ES and ET along the pseudo-binary line between the end members. For the GST based compounds, separate calculations have been performed for the metastable rock salt structure to determine the ES and ET values.

## Data availability

The authors declare that the source data supporting the findings of this study are available within the paper and its supplementary information file, where Supplementary Table 1 contains the majority of data analyzed in the manuscript. Correspondence and other requests for materials or data related to this study should be addressed to the corresponding author and will be provided upon reasonable request.

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

## Acknowledgements

We acknowledge the computational resources granted from RWTH Aachen University under project RWTH0508. This work was supported in part by the Deutsche Forschungsgemeinschaft (SFB 917) and in part by the Federal Ministry of Education and Research (BMBF, Germany) in the project NEUROTEC (16ES1133 K). We also acknowledge the work of Carl-Friedrich Schön who helped with the calculation of ES/ET and Sophia Wahl who took part in the discussion on the link between sample reflectance and the minimum time for crystallization. H.T. acknowledges Grants-in-Aid for Scientific Research (A) (JP18H03675) and Specially Promoted Research (JP20H05619) from the Japan Society for the Promotion of Science (JSPS). The critical reading of the manuscript by Christophe Bichara is gratefully acknowledged.

## Author contributions

M.W. initiated the project upon discussions with P.F. and conceptualized it. Sample preparation as well as experimental work was organized and performed by C.P., M.M., J.P., A.Y., N.H., and P.K. The paper was written by M.W., C.P., M.M., and J.P. with contributions from H.T., S.W., E.V., and F.P. as well as support from all co-authors. All authors have given approval to the final version of the manuscript.

## Funding

## Competing interests

The authors declare no competing interests.
