## [Peer Review File · Nature Communications]

REVIEWER COMMENTS

Reviewer #1 (Remarks to the Author):

This manuscript presents interesting and timely work on crystallization in chalcogenide systems. The central concept is that the composition can be used to control the rate of crystallization, over several orders of magnitude. Therefore, the desired kinetics (needed, for example, for phase-change memory) should be obtainable by design. This concept is, of course, highly attractive.

The manuscript is exceptionally well written, with the motivation for the work and the methods all explained clearly. There is a good division of content between the main text and the Supplementary Information. Figure 4 is a particularly good presentation of the data.

The experimental methods are appropriate for the study. The reviewer particularly appreciated Section III in Supplementary Information, which explains the reason to study sputter-deposited films (despite the fact that these films do not closely represent the properties of the melt-quenched state relevant for memory device operation).

The authors claim (p. 4) that “the correlation between the reflectance of the amorphous film before crystallization and the minimum time for crystallization τ is less evident”, and in Supplementary Information (p. 6) this point is made more strongly: “We can hence conclude that the minimum crystallization time is not closely related to the reflectance of the amorphous samples, but instead is closely related to the reflectance of the crystalline samples.” As the authors note in the main text (p. 4), “this is surprising since crystallization should depend on the properties of both the amorphous and crystalline states”.

These points are important for the authors, as they suggest (main text, p. 4) that they “imply that the crystalline state and its electronic structure, which govern optical properties seem to have a dominant impact on crystallization for phase change materials. How can this finding be rationalized?”

The reviewer is not convinced by this claim that, roughly speaking, the crystalline state is more important than the amorphous state. The authors’ argument rests on the comparison between Fig. 2b (showing a linear variation of τ with the reflectance of the crystal) with Fig. S3 (showing a step-like dependence of τ on the reflectance of the amorphous phase). The discussion in Supplementary Information, in the four lines immediately below Fig. S3, seems misguided. It is often the case that the key parameter is one that takes the system from one mechanism to another, and that change of mechanism implies a step change. An example is the ductile-to-brittle transition as the temperature is lowered (or as the strain rate is increased): the transition is step-like, and definitely not gradual. Another example (more closely connected to the present case) is where crystal growth rates as a function of undercooling in metallic alloy liquids show steps at the onset of solute trapping or disorder trapping.

From the reviewer’s perspective, and contrary to the authors’ view, Fig. S3 provides clear evidence for the IMPORTANCE of the amorphous state. The reviewer would argue that Fig. S3 shows a sharp transition when the reflectance of the amorphous phase is about 0.14. Above that value, the phase is more metallic, and the crystallization is faster; below that value, the phase is covalent and the crystallization is much slower. (We should note, however, that the phase from within which crystallization occurs is the liquid, not the solid amorphous phase, and that there might be a covalent-to-metallic transition on heating the amorphous phase.) From that perspective, the reflectivity of the crystalline phase could be of secondary importance.

Independent of any detail, it is clear that at the crystal-liquid interface, the preponderant mobility is in the liquid. Thus it will always be natural to link the rate of crystallization (i) to the atomic mobility in the liquid, and (ii) to the changes in structure (possibly related to changes in electronic state) at the

crystal-liquid interface, but not to the crystal itself.

In the Supplementary Information, there are repeated typographical errors ('crystallitization' instead of 'crystallization'; 'quenched' instead of 'quenched'). More care is needed in the presentation of this material.

The reviewer is eager to see these results published, but feels strongly that the authors need to reconsider their conclusion on the relative importance of the crystalline and amorphous states!

Reviewer #2 (Remarks to the Author):

The authors have studied three series of PCMs: GeTe-GeSe, GeTe-SnTe, and GeTe-Sb₂Te₃ to establish the relationships between crystallization and verification kinetics and bonding parameters, that is, the number of shared and transferred electrons between the neighboring atoms. The used methods, a pump laser setup and conventional and/or flash DSC, allow reliable results to be obtained also combined with quantum chemical calculations. The quality of experiments and statistical analysis is good.

It was found that increasing covalency on tellurium substitution by Se slows down the characteristic crystallization time by 4 orders of magnitude, while germanium substitution by Sn reduces the crystallization time by a factor of 25. The crystallization time correlates with the number of electrons shared (ES) and anti-correlates with optical reflectance of the crystalline sample. The vitrification kinetics was found to follow the Turnbull criterion, the reduced glass transition temperature, $T_{rg} = T_g/T_m$, or its related quantity, the reduced onset temperature, $T_{ro} = T_o/T_m$, for real PCMs. The T_{ro} also increases with ES, thus suggesting that crystalline and amorphous phases and their properties are equally dependent on specific bonding parameters. In other words, the authors have found a close relationship between crystallization and vitrification for several chalcogenides. This finding is not new and the authors have emphasized that similar relations were reported more than 50 years ago. However, the used efficient and reliable experimental methods and correlations with chemical bonding parameters are novel and might be of interest for broad readership.

Nevertheless, a striking exception was observed for GST alloys; they are characterized by fast crystallization and simultaneously by increased stability of the amorphous phase. This discrepancy was not discussed properly limited to a single sentence "It would be rewarding to explain the increased glass-forming ability of Ge₂Sb₂Te₅ compared to GeTe, which might be related to a more pronounced Peierls distortion of amorphous Ge₂Sb₂Te₅." A more detailed and well-reasoned discussion seems to be necessary for otherwise convincing and well-organized and written research.

Reviewer #3 (Remarks to the Author):

The manuscript is ostensibly about the design rules for crystallization and vitrification kinetics of promising chalcogenide-based phase-change alloys. The crystallisation and properties of those materials have been studied extensively since their discovery (including many theoretical and experimental works by Prof. Wuttig). This seems to me an extremely interesting and important for application (since the representatives from the Micron are the co-authors) paper containing a rich of important theoretical and experimental data. Although I am not able to assess the theoretical aspects (quantum chemical calculations) of the manuscript in full details, I can certainly admit, that analysis can be useful for designing and tailoring the crystallisation of single-phase phase change alloys in particular and other similar alloys (e.g. phase-change heterostructures) in general. However, here are the aspects of the paper that were quite confusing to me:

1. I have some concerns on the measurements of crystallisation times. The crystallisation time of GeTe was measured by the authors to be 614 ns. Other experimental results report the time of 60 ns

(see in Adv. Optical Mater. 5 (2017) 1700169). The authors should explain in the manuscript this difference and add more discussion on available data for crystallisation speed of other compositions (e.g. the reported crystallization time for Ge₂Sb₂Te₅ is in the order of 30 ns as reported in ACS Appl. Mater. Interfaces 11 (2019) 41544–41550).

2. More details of pump-probe experiment (e.g. pulse durations, pulse numbers, type and rise/fall time of photodetector etc). should be provided by the authors.

3. The authors used a 100 nm top layer of (ZnS)₈₀(SiO₂)₂₀ in their experiments. What was the impact of this layer on crystallisation time?

4. I recommend the authors to use fluency units instead of pulse power.

Reply to Reviewer #1:

Dear Reviewer #1,

Thank you very much for your thoughtful comments and suggestions. We are very happy about your recommendation to publish our manuscript and truly appreciate your many insightful comments and criticism. Below, we address your review sequentially, citing the report in full in *italics*, and interspersing our point-by-point **responses**.

Reviewer #1:

This manuscript presents interesting and timely work on crystallization in chalcogenide systems. The central concept is that the composition can be used to control the rate of crystallization, over several orders of magnitude. Therefore, the desired kinetics (needed, for example, for phase-change memory) should be obtainable by design. This concept is, of course, highly attractive.

The manuscript is exceptionally well written, with the motivation for the work and the methods all explained clearly. There is a good division of content between the main text and the Supplementary Information. Figure 4 is a particularly good presentation of the data.

The experimental methods are appropriate for the study. The reviewer particularly appreciated Section III in Supplementary Information, which explains the reason to study sputter-deposited films (despite the fact that these films do not closely represent the properties of the melt-quenched state relevant for memory device operation).

Response:

We really appreciate these kind and thoughtful words.

The authors claim (p. 4) that “the correlation between the reflectance of the amorphous film before crystallization and the minimum time for crystallization τ is less evident”, and in Supplementary Information (p. 6) this point is made more strongly: “We can hence conclude that the minimum crystallization time is not closely related to the reflectance of the amorphous samples, but instead is closely related to the reflectance of the crystalline samples.” As the authors note in the main text (p. 4), “this is surprising since crystallization should depend on the properties of both the amorphous and crystalline states”.

These points are important for the authors, as they suggest (main text, p. 4) that they “imply that the crystalline state and its electronic structure, which govern optical properties seem to have a dominant impact on crystallization for phase change materials. How can this finding be rationalized?”

The reviewer is not convinced by this claim that, roughly speaking, the crystalline state is more important than the amorphous state. The authors’ argument rests on the comparison between Fig. 2b (showing a linear variation of τ with the reflectance of the crystal) with Fig. S3 (showing a step-like dependence of τ on the reflectance of the amorphous phase). The discussion in Supplementary Information, in the four lines immediately below Fig. S3, seems misguided. It is often the case that the key parameter is one that takes the system from one mechanism to another, and that change of mechanism implies a step change. An example is the ductile-to-brittle transition as the temperature is lowered (or as the strain rate

is increased): the transition is step-like, and definitely not gradual. Another example (more closely connected to the present case) is where crystal growth rates as a function of undercooling in metallic alloy liquids show steps at the onset of solute trapping or disorder trapping.

From the reviewer's perspective, and contrary to the authors' view, Fig. S3 provides clear evidence for the IMPORTANCE of the amorphous state. The reviewer would argue that Fig. S3 shows a sharp transition when the reflectance of the amorphous phase is about 0.14. Above that value, the phase is more metallic, and the crystallization is faster; below that value, the phase is covalent and the crystallization is much slower. (We should note, however, that the phase from within which crystallization occurs is the liquid, not the solid amorphous phase, and that there might be a covalent-to-metallic transition on heating the amorphous phase.) From that perspective, the reflectivity of the crystalline phase could be of secondary importance.

Independent of any detail, it is clear that at the crystal-liquid interface, the preponderant mobility is in the liquid. Thus it will always be natural to link the rate of crystallization (i) to the atomic mobility in the liquid, and (ii) to the changes in structure (possibly related to changes in electronic state) at the crystal-liquid interface, but not to the crystal itself.

Response:

We are very impressed by your detailed review and the knowledgeable comments you provide, commenting on interesting phenomena such as the step change in the ductile-to-brittle transition or the impact of solute trapping on crystal growth rates in undercooled liquids of metallic alloys. It would be a pleasure to discuss with you in person how to develop a more coherent view on nucleation and growth in phase change materials.

As input for such a discussion and in an attempt to improve the clarity of our statements, we can summarize the present state of our understanding as follows:

- a) We agree with you that the decisive factors that govern crystallization are the thermodynamic driving force for crystallization, the interfacial tension between crystal and glass (undercooled liquid) and the atomic mobility in the undercooled liquid.
- b) So far, we have not seen any evidence that classical nucleation and growth theory fail to describe the corresponding processes in PCMSs. Yet, phase change materials show a few peculiarities which are remarkable. The atomic arrangement in the crystalline phase differs significantly, even as far as the short range order is concerned, from the atomic arrangement in the amorphous [Nature Materials 3, 703 (2004)] and undercooled liquid state [Science 364, 1062 (2019)]. Yet, the interfacial energy is surprisingly small [J. Appl. Physics 98, 54910 (2005)]. This is a rather unconventional combination of properties. The difference in atomic arrangement (and chemical bonding) leads to rather different properties of both phases [Nature Materials 7, 653 (2008)], while the rather modest interfacial energy supports fast nucleation.
- c) It is also important that the atomic mobility is high in the undercooled liquid state, a point that you stress in your report. This is apparently facilitated by a liquid – liquid phase transition which has recently been observed by time-resolved x-ray diffraction [Science 364, 1062 (2019)]. Upon cooling down the liquid phase, a transition to an undercooled liquid phase with Peierls distortion is observed. This change in atomic arrangement could also

explain the fragile to strong transition which is observed in the viscosity of several phase change materials [e.g. Chemistry of Materials 27, 5641 (2015)].

- d) The atomic arrangement in amorphous phase change materials is presumably also predominantly octahedral-like, but with a more pronounced Peierls-like distortion [Nature Communications 6, 7467 (2015), Phys. Rev. B 81, 081204(R) (2010)]. This also explains the pronounced difference between the atomic arrangement of amorphous and crystalline phase change materials like $\text{Ge}_2\text{Sb}_2\text{Te}_5$, as already reported in 2004 by Kolobov et al. [Nature Materials 3, 703 (2004)] This raises the question, which chalcogenides show such a pronounced difference in atomic arrangement and physical properties between the amorphous and crystalline state. Interestingly, there is a well-defined border as we have shown in a manuscript which is presently reviewed by Advanced Materials with favorable comments by the referees [arXiv:2008.10219]. This conclusion is important for the present discussion, since the map we show in figure 4 is a map for the crystalline phases. All amorphous phases are characterized by ES values in the covalent region of the map [unpublished data, manuscript is in preparation]. This is consistent with the experimental data which show a smooth transition from GeTe to GeSe, while the properties of the crystalline phase show a step change upon going from GeTe to GeSe, as observed by discontinuous changes of the dielectric function and the optical dielectric constant, the Born effective charge and the bond breaking [arXiv:2008.10219]. Hence, we are convinced that the crystalline state governs the crystallization kinetics, but are aware that this conclusion is counter-intuitive.

Changes made:

We have tried to provide a more thorough discussion of the impact of the amorphous and crystalline phase on crystallization kinetics. In particular, we add several sentences regarding the atomic mobility of the (undercooled) liquid phase and the importance of the Peierls distortion in the undercooled liquid phase.

*In theories discussing crystallization kinetics⁹, the **three** fundamental quantities are the heat of fusion (crystallization), i.e. the thermodynamic driving force for crystallization, the interfacial tension between crystal and glass, a measure of the similarity of both phases^{7,8} **as well as the atomic mobility in the undercooled liquid phase.***

Indeed, it has recently been shown that the undercooled liquid of phase materials like $\text{Ge}_{15}\text{Sb}_{85}$ and $\text{Ag}_4\text{In}_3\text{Sb}_{67}\text{Te}_{26}$ are characterized by a liquid – liquid phase transition, which marks the onset of Peierls distortions at lower temperatures [Science 364, 1062 (2019)]. Presumably this phase transition provides the rapid atomic rearrangement required for fast crystallization at elevated temperatures, yet stability against recrystallization at lower temperatures.

In the supplement we also provide further arguments why we believe that the crystalline phase is more important than the amorphous phase regarding crystallization kinetics. Yet, we also try to phrase our text in a way that the reader can notice that we are expressing a strong belief, which

should be supported and validated by further data. We are currently also analyzing the bonding in amorphous phase change materials but will need another month or two before submission.

*This aspect has recently been investigated in more detail for a series of samples along the pseudo-binary line between GeTe and GeSe, as well as two other similar lines [arXiv:2008.10219]. This study has shown unequivocally that the optical properties hardly change upon the transition from amorphous GeTe to amorphous GeSe. Instead, a much more pronounced change of optical properties has been observed for the crystalline samples along the GeTe-GeSe pseudo-binary line. Increasing the Ge content leads to a monotonous decrease of the optical dielectric constant ϵ_{∞} , which can be attributed to an increase of the Peierls distortion in the crystalline phase. At a critical Se concentration of around 60%, two different crystalline phases are observed which differ significantly in optical reflectivity and optical dielectric constant ϵ_{∞} . These two different samples also differ in their degree of Peierls distortion [arXiv:2008.10219] This indicates that there are much more pronounced changes of the bonding and structure and hence also physical properties along the **crystalline** GeTe-GeSe pseudo-binary line than along the **amorphous** GeTe-GeSe pseudo-binary line. Apparently, the changes in optical properties of the crystalline samples are thus linked to changes in crystallization kinetics. While this conclusion appears reasonable, further data such as quantum-chemical calculations to characterize the bonding in amorphous phase change materials are needed to quantify trends for the bonding in this phase with stoichiometry as well.*

In the Supplementary Information, there are repeated typographical errors ('crystallitization' instead of 'crystallization'; 'quenched' instead of 'quenched'). More care is needed in the presentation of this material.

Response:

We apologize for the typographical errors in the Supplementary Information and have corrected these errors.

Changes made:

The errors in the Supplementary Information have been corrected:

The reviewer is eager to see these results published, but feels strongly that the authors need to reconsider their conclusion on the relative importance of the crystalline and amorphous states!

Response:

As outlined above, we have reconsidered our arguments and now provide further evidence, why we believe that the conclusion is correct, but also mention more clearly that we express a strong belief, which needs additional data for its validation. Hence, we hope that this revised presentation

together with the arguments provided make the manuscript acceptable for publication in its present form.

Reply to Reviewer #2:

Dear Reviewer #2,

Thank you very much for your helpful comments and suggestions. We are very happy about your recommendation to publish our manuscript and appreciate your comment to extend our discussion of one crucial observation. Below, we address your review sequentially, citing the report in full in *italics*, and interspersing our point-by-point **responses**.

Reviewer #2:

The authors have studied three series of PCMs: GeTe-GeSe, GeTe-SnTe, and GeTe-Sb₂Te₃ to establish the relationships between crystallization and vitrification kinetics and bonding parameters, that is, the number of shared and transferred electrons between the neighboring atoms. The used methods, a pump laser setup and conventional and/or flash DSC, allow reliable results to be obtained also combined with quantum chemical calculations. The quality of experiments and statistical analysis is good. It was found that increasing covalency on tellurium substitution by Se slows down the characteristic crystallization time by 4 orders of magnitude, while germanium substitution by Sn reduces the crystallization time by a factor of 25. The crystallization time correlates with the number of electrons shared (ES) and anti-correlates with optical reflectance of the crystalline sample. The vitrification kinetics was found to follow the Turnbull criterion, the reduced glass transition temperature, $Trg = Tg/Tm$, or its related quantity, the reduced onset temperature, $Tro = To/Tm$, for real PCMs. The Tro also increases with ES, thus suggesting that crystalline and amorphous phases and their properties are equally dependent on specific bonding parameters. In other words, the authors have found a close relationship between crystallization and vitrification for several chalcogenides. This finding is not new and the authors have emphasized that similar relations were reported more than 50 years ago. However, the used efficient and reliable experimental methods and correlations with chemical bonding parameters are novel and might be of interest for broad readership.

Response:

We really appreciate your favorable evaluation and recommendation of our manuscript.

Nevertheless, a striking exception was observed for GST alloys; they are characterized by fast crystallization and simultaneously by increased stability of the amorphous phase. This discrepancy was not discussed properly limited to a single sentence "It would be rewarding to explain the increased glass-forming ability of Ge₂Sb₂Te₅ compared to GeTe, which might be related to a more pronounced Peierls distortion of amorphous Ge₂Sb₂Te₅." A more detailed and well-reasoned discussion seems to be necessary for otherwise convincing and well-organized and written research.

Response:

You raise an important point in suggesting to extend our discussion on the stability of amorphous GST alloys. Indeed, providing only one sentence to discuss the increased stability of amorphous GST alloys is rather short. Yet, we have tried to refrain from becoming too speculative in our manuscript. To account for your suggestion, we now provide a more detailed discussion of the relevance of the crystalline phase for crystallization kinetics (to account for the comments of reviewer '1), but also

extend the discussion of the relevance of the Peierls distortion for amorphous phase change materials and how this might stabilize the amorphous phase of GST alloys.

Changes made:

We thus added the following sentence in the manuscript,

It would be rewarding to explain the increased glass-forming ability of $\text{Ge}_2\text{Sb}_2\text{Te}_5$ compared to GeTe , which might be related to a more pronounced Peierls distortion of amorphous $\text{Ge}_2\text{Sb}_2\text{Te}_5$ as discussed in more detail in the supplement.

and extended the discussion considerably in the supplement:

In this context, the finding reported in figure 3.b. also raises questions. There it was shown that amorphous GST alloys are characterized by an increased onset temperature T_{ro} for glass formation. One can wonder if this is due to an increased Peierls distortion in amorphous GST alloys caused by the additional disorder in ternary compounds. This increased disorder could facilitate larger Peierls distortions and thus an increased stability of the amorphous phase against crystallization. Yet, at elevated temperatures we expect a fragile to strong transition of the viscosity. Yet, to validate this speculation a more detailed analysis of the atomic arrangement and the dielectric function, as well as the temperature dependence of the viscosity would be required to turn this plausible hypothesis into a proven fact.

Reply to Reviewer #3:

Dear Reviewer #3,

Thank you very much for your helpful comments and suggestions. We are very happy to read that our ‘analysis can be useful to design and tailor phase change materials’. Below, we address your review sequentially, citing the report in full in *italics*, and interspersing our point-by-point **responses**.

Reviewer #3:

The manuscript is ostensibly about the design rules for crystallization and vitrification kinetics of promising chalcogenide-based phase-change alloys. The crystallisation and properties of those materials have been studied extensively since their discovery (including many theoretical and experimental works by Prof. Wuttig). This seems to me an extremely interesting and important for application (since the representatives from the Micron are the co-authors) paper containing a rich of important theoretical and experimental data. Although I am not able to assess the theoretical aspects (quantum chemical calculations) of the manuscript in full details, I can certainly admit, that analysis can be useful for designing and tailoring the crystallisation of single-phase phase change alloys in particular and other similar alloys (e.g. phase-change heterostructures) in general. However, here are the aspects of the paper that were quite confusing to me:

1. I have some concerns on the measurements of crystallisation times. The crystallisation time of GeTe was measured by the authors to be 614 ns. Other experimental results report the time of 60 ns (see in Adv. Optical Mater. 5 (2017) 1700169). The authors should explain in the manuscript this difference and add more discussion on available data for crystallisation speed of other compositions (e.g. the reported crystallization time for Ge₂Sb₂Te₅ is in the order of 30 ns as reported in ACS Appl. Mater. Interfaces 11 (2019) 41544–41550).

Response:

You address an important question here, the crystallization times measured by us and how they compare with other data reported in the literature. This aspect has also been mentioned by reviewer ‘1, who commented on the usefulness of our explanation in Section III of the Supplementary Information, where we discuss, why we studied amorphous films in their as-deposited state. Studies of crystallization using an optical tester, as employed here, differ in two important aspects: do the authors study re-crystallization of melt-quenched samples or as-deposited films and/or do they study re-crystallization of capped or un-capped films. We now also add a discussion on the reasons why we capped the films in Section III of the Supplementary Information.

Changes made:

We have added the following text to the supplement:

Finally, we have employed a capping layer to protect the phase change film from oxidation. Previous studies have shown that exposure to air leads to the formation of a film of GeO_2 on top of the phase change film [Journal of Applied Physics **96**, 5557 (2004)]. This leads to a loss of Ge in the underlying phase change film, which changes the stoichiometry of the film and lowers the activation barrier for crystallization and thus leads to significantly shorter crystallization times. To be able to unequivocally relate the film stoichiometry to crystallization speed, the use of such a protective layer is mandatory. The resulting layer stack also resembles the stack employed in optical and electrical phase change memories.

2. More details of pump-probe experiment (e.g. pulse durations, pulse numbers, type and rise/fall time of photodetector etc.) should be provided by the authors.

Response:

We provide more information on the laser set-up employed to perform the studies of crystallization kinetics. We should stress that we used a pump – probe set-up, where two different laser wave-lengths were used to modify and characterize the samples, utilizing wave-lengths of 658 and 639 nm. To measure the energy of the reflected probe laser, a slow photodetector is utilized to improve the signal to noise ratio for the change of sample reflectivity, and an avalanche photodiode detector (APD detector) featuring a bandwidth of 1.2 GHz generates an amplified signal to be monitored by the oscilloscope to characterize the pump pulse.

Changes made:

We have added the following information to the text:

“The probe laser measured the change in reflectance after each single pump pulse compared to the reflectance before the pump pulse, providing a PTE-diagram, which depicts the effect (E) of different pulse power densities (P) between 0 and 90 mW and pulse lengths (T) between 10^1 ns and 10^8 ns (see figure S6)”.

Furthermore, we added the following information in the supplement:

“When a rectangular laser pulse hits the sample, the absorbed energy results in local heating of the illuminated area. Depending on the applied pulse power and pulse length, the sample can be switched between its amorphous and its crystalline phase. The change in the dielectric function upon switching results in a change of sample reflectance measured by the probe laser before and after the pump pulse.”

3. The authors used a 100 nm top layer of $(\text{ZnS})_{80}(\text{SiO}_2)_{20}$ in their experiments. What was the impact of this layer on crystallisation time?

Response:

We have addressed this point already in our comparison with other studies of crystallization times (see answer to question 1).

4. I recommend the authors to use fluency units instead of pulse power.

Response:

We have addressed this point in the manuscript. Indeed, in many experiments optical excitation is studied as a function of fluency. Yet, in these studies often the pulse length is not varied, in contrast to the experiments reported here. We could instead compare different experiments based on the laser power per sample area. Yet, this quantity depends on details of the optical set-up such as the quality of the laser focus. We have spent a significant amount of time to ensure that the pulse power that reaches the sample does not fluctuate from pulse to pulse, as can also be seen by the low set-up noise in the PTE diagrams, yet, we want to refrain from using the laser power per sample area throughout the text, since the precise number of this quantity depends upon a very precise quantification of the spatial profile of our laser beam, while we have focused on the laser pulses being highly reproducible in terms of their pulse powers and pulse lengths.

Changes made: We thus added the following passage in the methods section:

“The maximum laser power corresponds to a power density of 4.3×10^6 W/cm² in the center of the beam.”

And we added the following passage in the Supplementary Information section VII:

“The effective central power density of the laser E is given by:

$$E = \frac{8}{\pi} \cdot \frac{P}{d^2},$$

where d is the diameter ($1/e^2$) of the laser beam of $2.3 \mu\text{m}$.”

REVIEWER COMMENTS

Reviewer #1 (Remarks to the Author):

The authors have made useful revisions to the manuscript to explain their reasoning more clearly. In their responses, the authors state that "the crystalline state governs the crystallization kinetics". The reviewer still feels that this narrow view is not justified. It is clear that for the different compositions that characteristics of the amorphous/glassy state, the characteristics of the crystal, and the vitrification and crystallization kinetics are all correlated in various ways. As the authors point out, Figure 3 (particularly Fig. 3a) shows a clear correlation between the bonding in the crystal and the stability of the glass. Thus in considering the link between the bonding in the crystal and the crystallization kinetics, it is not easy to distinguish between 'causality' and 'correlation'.

Perhaps the real question is not the role of the amorphous/glassy state, but rather the role of the liquid. The statement "the crystalline state governs the crystallization kinetics" seems to deny any role for the liquid, yet even the authors seem to assume that there is such a role. For example, in one of their additions to the revised text, they note:

"Indeed, it has recently been shown that the undercooled liquid of phase materials like Ge₁₅Sb₈₅ and Ag₄In₃Sb₆₇Te₂₆ are characterized by a liquid – liquid phase transition, which marks the onset of Peierls distortions at lower temperatures [Science 364, 1062 (2019)]. Presumably this phase transition provides the rapid atomic rearrangement required for fast crystallization at elevated temperatures, yet stability against recrystallization at lower temperatures." In that view, the nature of the liquid is important!

Importantly, however, the authors do not make the statement "the crystalline state governs the crystallization kinetics" in the manuscript itself. The revised wordings are mostly appropriately careful.

On lines 109-110, we find:

"the crystalline state and its electronic structure ... seem to have a dominant impact on crystallization for phase change materials" -- that seems fine.

On the other hand, on lines 262-263, we find:

"shows that the number of electrons shared governs the crystallization time τ ". That seems a bit too strong. A suggestion for replacement would be something like: "shows that the crystallization time τ is clearly correlated with the number of electrons shared". This can be regarded as an optional revision.

Reviewer #2 (Remarks to the Author):

The authors have fully addressed my comments related to the increased glass stability of Ge₂Sb₂Te₅ compared to GeTe. I recommend the revised version for publication.

Reviewer #3 (Remarks to the Author):

The authors did not address my first comment appropriately. After careful reading of the mentioned manuscript (Adv. Optical Mater. 5 (2017) 1700169), I found the following information regarding capping layer and a state of amorphous GeTe. The GeTe thin films in Adv. Optical Mater. 5 (2017) 1700169 were protected by a capping layer from the oxidation (see in Experimental section) and the authors used GeTe thin films in as-deposited state (read section 2.4). Thus, experimental conditions were similar to the submitted manuscript. The only difference is a method of thin film growth. In the Adv. Optical Mater. the GeTe thin films were grown by pulsed laser deposition while in the submitted

paper the thin films were produced via DC-magnetron sputter deposition method. The kinetic energy of impinging atomic species in PLD is higher compared to magnetron (up to few hundred of eV in PLD vs a few tens of eV in thermal evaporation in sputtering-based methods). Thus, higher kinetic energy of PLD plasma can result in the formation of amorphous GeTe phase with better local ordering compared to sputter grown thin films. This is more plausible explanation, which should be mentioned in the submitted manuscript too (e.g. different ordering of as-deposited amorphous GeTe phase produced by other deposition methods might result in faster crystallization time). Otherwise the differences in crystallization time of GeTe (614 ns this work vs 60 ns reported in the literature) is difficult to explain, without appropriate discussion.

Regarding the reference Journal of Applied Physics 96, 5557 (2004) titled "Influence of Bi doping upon the phase change characteristics of Ge₂Sb₂Te₅": It assumes the formation of GeO₂ oxide layer on top of PCM as was written on page 5559 ".we have assumed that the oxide film on top will also contain phase change material". In the case of Ge₂Sb₂Te₅, the oxide layer consists of Ge, Sb and O species.

Reply to Reviewer #1:

Dear Reviewer #1,

Thank you very much for your thoughtful comments and suggestions. We are happy to hear that you appreciate the revisions we made and suggest an optional revision. Below, we address your review sequentially, citing the report in full in *italics*, and interspersing our point-by-point **responses**.

Reviewer #1 (Remarks to the Author):

The authors have made useful revisions to the manuscript to explain their reasoning more clearly. In their responses, the authors state that “the crystalline state governs the crystallization kinetics”. The reviewer still feels that this narrow view is not justified. It is clear that for the different compositions that characteristics of the amorphous/glassy state, the characteristics of the crystal, and the vitrification and crystallization kinetics are all correlated in various ways. As the authors point out, Figure 3 (particularly Fig. 3a) shows a clear correlation between the bonding in the crystal and the stability of the glass. Thus in considering the link between the bonding in the crystal and the crystallization kinetics, it is not easy to distinguish between ‘causality’ and ‘correlation’.

Perhaps the real question is not the role of the amorphous/glassy state, but rather the role of the liquid. The statement “the crystalline state governs the crystallization kinetics” seems to deny any role for the liquid, yet even the authors seem to assume that there is such a role. For example, in one of their additions to the revised text, they note:

“Indeed, it has recently been shown that the undercooled liquid of phase materials like Ge₁₅Sb₈₅ and Ag₄In₃Sb₆₇Te₂₆ are characterized by a liquid – liquid phase transition, which marks the onset of Peierls distortions at lower temperatures [Science 364, 1062 (2019)]. Presumably this phase transition provides the rapid atomic rearrangement required for fast crystallization at elevated temperatures, yet stability against recrystallization at lower temperatures.” In that view, the nature of the liquid is important!

Importantly, however, the authors do not make the statement “the crystalline state governs the crystallization kinetics” in the manuscript itself. The revised wordings are mostly appropriately careful.

On lines 109-110, we find:

“the crystalline state and its electronic structure ... seem to have a dominant impact on crystallization for phase change materials” -- that seems fine.

On the other hand, on lines 262-263, we find:

“shows that the number of electrons shared governs the crystallization time τ ”. That seems a bit too strong. A suggestion for replacement would be something like: “shows that the crystallization time τ

is clearly correlated with the number of electrons shared". This can be regarded as an optional revision.

Response:

We appreciate that you distinguish between the statements we make in our manuscript and the belief expressed in our reply to you. Specifically, you say, that the *revised wordings are mostly appropriately careful.*

Changes made:

We changed the annotated sentence as you suggested.

*Dependence of the minimum crystallization time τ a) and the reduced onset temperature T_{ro} for glass formation b) upon two chemical bond quantifiers, the number of electrons transferred and shared between adjacent atoms. A pronounced decrease of the minimum time to crystallize and the glass-forming ability is observed in the metavalent bonding region (green background) between covalent (red) and metallic bonding. Figure 4a, in conjunction with Figure 2a, *shows that the number of electrons shared governs the crystallization time τ shows that the crystallization time τ is clearly correlated with the number of electrons shared.* For the Se-rich compounds, which are covalently bonded such as GeSe, crystallization is not even discernible in the optical tester.*

Additional Response:

You mention specifically that *considering the link between the bonding in the crystal and the crystallization kinetics, it is not easy to distinguish between 'causality' and 'correlation'.* We fully agree with your statement. In this manuscript we demonstrate that there is a correlation between crystallization kinetics and the properties of the crystalline state, a finding which intrigues both you and us, since you comment in the last round of the review 'I want to see this manuscript published'. Crystallization, i.e. the transition between the amorphous (undercooled liquid) and crystalline state, must depend upon the properties of both the amorphous (undercooled liquid) and crystalline state. This is a fact that we are both aware of. The reason, why we believe that for phase change materials the crystalline state is more important and thus dominates crystallization kinetics, comes from studies (accepted for publication by Advanced Materials two days ago), which are beyond the scope of the present manuscript and hence are not discussed here. We have recently spent significant efforts to unravel systematic property trends (for a large number of chalcogenides). These studies reveal that there are distinct changes of properties and bonding for crystalline chalcogenides upon isoelectronic replacement, i.e. going from GeTe to GeSe and from Sb₂Te₃ to Sb₂Se₃. For the amorphous materials, on the contrary, a smooth and continuous property change has been observed. Hence, we did not find a smoking gun for fast crystallization kinetics based on properties and bonding in **amorphous** chalcogenides. You mention in your reply that the behavior in the (under-cooled) liquid state might be telling, i.e. the question, if there is a liquid – liquid transition. This is indeed a very interesting hypothesis. Yet, we believe that such a transition in chalcogenides is presumably related to the prevalence of metavalent bonding in the crystalline state, a belief which requires careful experimental and theoretical proof.

Reply to Reviewer #2:

Dear Reviewer #2,

We are happy to read that you recommend our revised manuscript for publication.

Reply to Reviewer #3:

Dear Reviewer #3,

Thank you very much for your thoughtful comments and careful suggestions. Below, we address your review sequentially, citing the report in full in *italics*, and interspersing our point-by-point **responses**. We hope that the revised manuscript meets your expectations and can be published in its present form.

Reviewer #3 (Remarks to the Author):

The authors did not address my first comment appropriately. After careful reading of the mentioned manuscript (Adv. Optical Mater. 5 (2017) 1700169), I found the following information regarding capping layer and a state of amorphous GeTe. The GeTe thin films in Adv. Optical Mater. 5 (2017) 1700169 were protected by a capping layer from the oxidation (see in Experimental section) and the authors used GeTe thin films in as-deposited state (read section 2.4). Thus, experimental conditions were similar to the submitted manuscript. The only difference is a method of thin film growth. In the Adv. Optical Mater. the GeTe thin films were grown by pulsed laser deposition while in the submitted paper the thin films were produced via DC-magnetron sputter deposition method. The kinetic energy of implying atomic species in PLD is higher compared to magnetron (up to few hundred of eV in PLD vs a few tens of eV in thermal evaporation in sputtering-based methods). Thus, higher kinetic energy of PLD plasma can result in the formation of amorphous GeTe phase with better local ordering compared to sputter grown thin films. This is more plausible explanation, which should be mentioned in the submitted manuscript too (e.g. different ordering of as-deposited amorphous GeTe phase produced by other deposition methods might result in faster crystallization time). Otherwise the differences in crystallization time of GeTe (614 ns this work vs 60 ns reported in the literature) is difficult to explain, without appropriate discussion.

Response:

Thank you very much for your careful review and comments. Indeed, we mistakenly assumed that in manuscript mentioned (Adv. Optical Mater. 5 (2017) 1700169) no capping layer was used. Hence, the only differences between this study and our investigation of crystallization in GeTe are:

- a) The absence of a heat barrier in the study mentioned above.
- b) The use of a much larger laser beam to crystallize the sample
- c) The difference in the deposition of GeTe, i.e. sputtering in our case and PLD in the study mentioned above

It is indeed plausible, as you suggest, that the most important difference is this difference in film deposition, which could lead to subtle differences in atomic arrangement. Indeed, such differences are known from studies of the effect of structural relaxation and priming, i.e. exciting the amorphous material by short laser pulses.

Changes made:

We thus removed the section about the relevance of capping layers.

Furthermore, the characterization of our samples with optical spectroscopy requires samples sizes of several mm, while we can only produce μm -sized regions of the melt-quenched state. Hence, we can only characterize the as-deposited amorphous state with our experimental tools, which are required to relate material properties to crystallization kinetics. Finally, we are presently unable to produce the necessary amounts of melt-quenched material to perform calorimetry measurements, which require about (0.1-2) μg of the material per measurement. Since we rely on laser quenching to produce the melt-quenched phase, only thin layers (up to approximately 50 nm in thickness) can be prepared due to the high absorption of the materials under investigation. Using thicker layers would result in an unsuccessful quenching process since too much heat is absorbed in the top area of the thin film while the rest of the film would hinder heat flow into the substrate, causing the molten area to crystallize rather than staying in the amorphous phase. Due to the limited power of the laser setup, the spot size cannot be increased without losing the ability to melt the sample at all. Thus, producing the amount of melt quenched material required for calorimetric measurements is not feasible. For these three reasons we have studied the crystallization of as-deposited amorphous chalcogenides.

~~*Finally, we have employed a capping layer to protect the phase change film from oxidation. Previous studies have shown that exposure to air leads to the formation of a film of GeO_2 on top of the phase change film. This leads to a loss of Ge in the underlying phase change film, which changes the stoichiometry of the film and lowers the activation barrier for crystallization and thus leads to significantly shorter crystallization times. To be able to unequivocally relate the film stoichiometry to crystallization speed, the use of such a protective layer is mandatory. The resulting layer stack also resembles the stack employed in optical and electrical phase change memories.*~~

Upon your suggestion, we have added a comment on the potential impact of different preparation methods.

The minimum time for crystallization differs between melt-quenched and as-deposited glassy states as has been shown by numerous studies³⁻⁵. In the last decade a number of reasons have been identified for this difference. Apparently, subcritical nuclei can be frozen in upon melt-quenching⁴. These nuclei presumably facilitate nucleation and hence speed up the crystallization process. Similar subcritical nuclei and thus a faster crystallization process can be realized upon deposition by other methods like pulsed

laser deposition due to the increased kinetic energy of the atoms compared to sputter deposition. At present, the study of the formation of such sub-critical nuclei is beyond the scope of ab-initio molecular dynamics simulations, which employ quenching times which are about 4 orders of magnitude shorter than the experimentally accessible quenching times. Hence, it seems that these computations provide a better model of the as-deposited state of sputtered samples. This is one reason why we focus on the as-deposited state here, since we plan to link the findings presented here to quantum-chemical calculations of the as-deposited amorphous state in the near future.

Regarding the reference Journal of Applied Physics 96, 5557 (2004) titled “Influence of Bi doping upon the phase change characteristics of Ge₂Sb₂Te₅”: It assumes the formation of GeO₂ oxide layer on top of PCM as was written on page 5559 “..we have assumed that the oxide film on top will also contain phase change material”. In the case of Ge₂Sb₂Te₅, the oxide layer consists of Ge, Sb and O species.

Response:

Based on your suggestion to focus on differences in the atomic arrangement of the as-deposited films, we have removed the part about the capping layer and its composition upon ambient oxidation.

Changes made:

We removed the section about the impact of a capping layer and the oxides formed upon ambient oxidation.

REVIEWERS' COMMENTS

Reviewer #1 (Remarks to the Author):

The authors have dealt satisfactorily with the points raised by the reviewers. This revised manuscript is recommended for publication without any need for further revision.

Reviewer #3 (Remarks to the Author):

The authors improved their manuscript significantly. Although I could not understand why the authors ignore the crystallization data for GeTe published in the discussed publication Adv. Optical Mater. 5 (2017) 1700169, I have no further comments on the manuscript and would recommend it for the publication.